



# An Online Ensemble Coupled Data Assimilation Capability for the Community Earth System Model: System Design and Evaluation

Jingzhe Sun[★,5], Yingjing Jiang[★,1,4], Shaoqing Zhang[*1,3,4], Weimin Zhang[*2], Lv Lu[1,4], Guangliang Liu[6], Yuhu Chen[3], Xiang Xing[2], Xiaopei Lin[1,3,4] and Lixin Wu[1,3,4]

[1]Key Laboratory of Physical Oceanography, Ministry of Education/Institute for Advanced Ocean Study/Frontiers Science Center for Deep Ocean Multispheres and Earth System (DOMES), Ocean University of China, Qingdao, China
[2]Southern Marine Science and Engineering Guangdong Laboratory, Zhuhai, China
[3]Qingdao Pilot National Laboratory for Marine Science and Technology (QNLM), Qingdao, China
[4]The College of Oceanic and Atmospheric Sciences, Ocean University of China, Qingdao, China
[5]Beijing Institute of Applied Meteorology, Beijing, China
[6]Shandong Provincial Key Laboratory of Computer Networks, Qilu University of Technology (Shandong Academy of Sciences), Jinan, China

[*]Corresponding author: S. Zhang(szhang@ouc.edu.cn); W. Zhang (wmzhang104@139.com)
[★] These authors contributed equally to this work.

**Abstract.** The Community Earth System Model (CESM) developed at the National Center of Atmospheric Research (NCAR) has been used worldwide for climate studies. This study extends the efforts of CESM development to include an online (i.e., in-core) ensemble coupled data assimilation system (CESM-ECDA) to enhance CESM's capability for climate predictability studies and prediction applications. The CESM-ECDA system consists of an online atmospheric data assimilation (ADA) component implemented to both the finite-volume and spectral-element dynamical cores, and an online oceanic data assimilation (ODA) component. In ADA, surface pressures (Ps) are assimilated, while in ODA, gridded sea surface temperature (SST) and ocean temperature and salinity profiles at real Argo locations are assimilated. The system has been evaluated within a perfect twin experiment framework, showing significantly reduced errors of the model atmosphere and ocean states through "observation"-constraints by ADA and ODA. The weakly CDA in which both the online ADA and ODA are conducted during the coupled model integration shows smaller errors of air-sea fluxes than the single ADA and ODA, facilitating the future utilization of cross-covariance between the atmosphere and ocean at the air-sea interface. A three-year CDA reanalysis experiment is also implemented by assimilating Ps, SST and ocean temperature and salinity profiles from the real world spanning the period 1978 to 1980 using 12 ensemble members. Results show that Ps RMSE is smaller than 20CR and SST RMSE is better than ERA-20C and close to CFSR. The success of the online CESM-ECDA system is the first step to implement a high-resolution long-term climate reanalysis once the algorithm efficiency is much improved.





## 1 Introduction

The Community Earth System Model (CESM) is a fully coupled global Earth system model developed by the National Center
for Atmospheric Research (NCAR), consisting of several geophysical component models (atmosphere, ocean, land, land ice,
sea ice, river runoff, and ocean wave) and a central infrastructure for coupling component models. It can simulate the past,
present, and future climate states of the Earth system (e.g., Chiodo et al., 2016; Glotfelty et al., 2017; Chang et al., 2020), and
has been used worldwide in numerous climate studies (e.g., Goldenson et al., 2012; Cheng et al., 2014; Gantt et al., 2014;
Fasullo & Nerem, 2016) to study the evolution mechanisms of climate and environment (e.g., Goosse & Holland, 2005; Bitz,
2008; Chandan & Peltier, 2018), the impact of natural processes and human activities on climate change, and the prediction of
climate change (e.g., Coelho & Goddard, 2009; Arblaster et al., 2011; Asefi-Najafabady et al., 2018).

Developing coupled data assimilation (CDA) is an inevitable requirement to improve initialization, state estimation, and
prediction of coupled models with observations available in multiple components of the Earth system (Zhang et al., 2020b).
Traditionally, data assimilation (DA) is carried out independently in an uncoupled model, such as assimilating atmospheric
observations into an atmosphere component model or assimilating oceanic observations into an ocean component model, which
is referred to as uncoupled DA (Derber & Rosati, 1989; Rosati et al., 1997; Saha et al., 2006; Balmaseda & Anderson, 2009).
When the uncoupled DA is used to initialize the coupled model integration, it is necessary to first combine the single
component analyses obtained independently from the single component DA to form the coupled initial conditions for the
coupled model. With the wide application of coupled models in the study of weather and climate systems, the demand for
CDA is rising rapidly (Penny et al., 2017). CDA refers to the joint assimilation of observations in different component systems
and allows to transfer and exchange observational information among different components dynamically and statistically (e.g.,
Lu et al., 2015; Sluka et al., 2016; Zhang et al., 2020b). It has been shown that CDA is able to improve the interannual climate
prediction skills of coupled models (e.g., Collins, 2002; Zhang et al., 2010b). Therefore, CDA is considered as an effective
method for the initialization of multi-component coupled Earth system models and the production of coupled reanalysis (Zhang,
55 2011).

As described by Penny et al. (2017), CDA can be broadly categorized into two approaches: weakly CDA (WCDA) and strongly
CDA (SCDA). WCDA is defined by the fact that the coupling occurs during the forecast stage by using a coupled forecast
model as DA is ongoing. WCDA assimilates the observations into the corresponding single model component of the coupled
model system and then transfers the observational information to other model components through flux exchange of the
coupled model. The second approach, SCDA, uses the cross-covariance of model components to directly assimilate the
observation of an earth system component into other coupled model components. The advantage of SCDA is that observations
at a given time have instantaneous impacts across all components during all available analyses (Penny et al., 2017; Zhang et
al., 2020b). Due to the difficulties in obtaining a high signal-to-noise ratio of the covariance between model components (Han
et al., 2013), by now the WCDA is still the common choice for assimilating observations into coupled models (e.g., Laloyaux



et al., 2016; Browne et al., 2019; Skachko et al., 2019; Tang et al., 2020; Mu et al., 2020) and in the meanwhile some studies have also discussed the SCDA (e.g., Negar et al., 2020). In this study, we use WCDA.

According to the mode of data transfer between the numerical model and assimilation algorithm, ensemble-based CDA can be divided into two categories: offline CDA and online CDA. In offline CDA, data are transferred between the model ensemble and assimilation algorithm through data reading/writing. The data assimilation research testbed (DART, Anderson et al., 2009)

developed by NCAR and the Gridpoint Statistical Interpolation (GSI) ensemble Kalman filter (EnKF) system (Kleist et al., 2009) released by the Developmental Testbed Center (DTC) are both based on an offline mode. Although the offline CDA is a convenient way to implement a CDA procedure on a relatively short scale, and the majority of the I/O cost in DART is low compared to the total time (Karspeck et al., 2018), an online CDA system is more efficient and necessary for climate studies. The online CDA is usually realized by integrating a numerical model and DA algorithm into one executable program, so that

data exchange between model and assimilation is generally realized by memory management. For example, the ensemble CDA (ECDA) system developed by Zhang et al. (2005 & 2007) based on the Geophysical Fluid Dynamics Laboratory (GFDL) coupled climate model and the Parallel Data Assimilation Framework (PDAF) developed by Nerger et al. (2005 & 2013) based on the fully coupled Alfred Wegener Institute, Helmholtz Center for Polar and Marine Research Climate Model (Mu et al., 2019; Nerger et al., 2020) are online CDA systems.

Extensive offline DA works with CESM have been explored in previous studies (e.g., Anderson et al., 2009; Raeder et al., 2012; Karspeck et al., 2013 & 2018). All these studies are based on an offline DA framework which needs to read and write restart files at every assimilating timestep. This time-consuming way makes it difficult to produce a long-term climate reanalysis. In order to develop coupled prediction and climate reanalysis, especially toward the goal of high-resolution CESM (CESM-HR, Zhang et al., 2020a) CDA for HR coupled model prediction initialization and reanalysis, a continuous CDA

within CESM is highly desirable, although the DART exists and the next-generation DA system (Joint Center for Satellite Data Assimilation JEDI DA system) is designed and has been developing to replace the GSI and other US agency DA systems. This paper serves as a documentation for the initial step of CESM-HR CDA development, the CESM-ECDA system design and its performance.

In the filtering algorithm, the DA method used in this study (which is ensemble adjustment Kalman filter, EAKF) is the same

as previous studies (Zhang & Anderson, 2003; Anderson et al., 2009; Karspeck et al., 2018), but we implement the CESM-ECDA as an online CDA system which uses the computer memory management to realize compiling the assimilation codes and the model codes into an executable file. Avoiding frequent huge data reading/writing is extremely important to realize efficient climate reanalysis especially for high resolution model cases. Although a lot of CESM DA work using DART has been done and a 12-year reanalysis has been presented (Karspeck et al., 2018), longer-term climate reanalysis is still a challenge.

Our motivation is to develop an online CESM-ECDA system that can support long-time integration and assimilation in the near future. By that, we can make a series of long-time scale (such as 40-year, or even 100-year) climate reanalysis experiments in different resolutions for climate assessment. This study is different from Karspeck et al. (2018) in the following two aspects:



1) The CESM-ECDA system passes data through memory instead of files, 2) The online DA system within SE dynamic-core atmosphere model can support high-resolution simulation.

This paper is organized as follows. Section 2 describes the CGCMs (Coupled General Circulation Model) used in this work, the ensemble filtering algorithm, and the perfect twin experiment design for the evaluation. The implementation of the ensemble-based online CESM DA capability is described in section 3. More specifically, section 3.1 describes the ADA system within the Community Atmosphere Model (CAM) model using the finite-volume dynamical core (dynamic-core; hereafter CAM-FV), section 3.2 describes the ADA system within CAM using the spectral-element dynamic-core CAM (hereafter

CAM-SE), section 3.3 discusses the ocean DA (ODA) system with the Parallel Ocean Program (POP) model. Section 4 describes the establishment and evaluation of online CESM-ECDA system. Finally, summary and discussions are given in section 5.

## 2 Model and Filtering Algorithm

### 2.1 Brief Description of CESM

The version of CESM used in this work is not the latest version of CESM2 (Danabasoglu et al., 2020), but based on an earlier version tagged as CESM1.3-beta17_sehires38, which is a version specifically developed to better support high-resolution CESM simulations (Small et al., 2014; Chang et al., 2020). The corresponding component model versions are the CAM version 5 (CAM5; Neale et al., 2012), the POP version 2 (POP2; Smith et al., 2010), the Community Ice Code version 4 (CICE4; Hunke & Lipscomb, 2008), and the Community Land Model version 4 (CLM4; Lawrence et al., 2011).

### 2.2 Brief Summary of EAKF

The ensemble adjustment Kalman filter (EAKF) was first designed by Anderson (2001) followed by some modifications (Anderson, 2003) and immediately applied to comprehensive general circulation models (Zhang & Anderson, 2003; Zhang et al., 2005 & 2007). As a popular variant of the traditional ensemble Kalman filter (EnKF; Evensen, 1994 & 2003), the EAKF has been widely used in DA research and applications. Compared with the traditional EnKF that randomly perturbs the

observations, the advantage of EAKF is that it can retain the higher-order moments (i.e., nonlinear information) in the prior samples (Zhang & Anderson, 2003). The EAKF does not need to perturb observations, thus avoiding the generating of extra noise into the analysis system. Based on a systematic analysis of the existing ensemble filtering algorithms, Tippett et al. (2003) pointed out that the EAKF method is one realization of the ensemble square root filter (EnSRF) and a deterministic filtering algorithm. When the error distribution of each scalar observation is independent of each other, a sequential filtering method

can be used to assimilate each scalar observation one by one. When the observation errors are correlated, the singular value decomposition (SVD) can be used to first decorrelate the errors, and then the sequential filtering can be carried out.

As a sequential filter, the EAKF can be conveniently implemented by two steps consisting of two key equations (Anderson, 2001 & 2003; Zhang & Rosati, 2010a). The first equation computes the observational increment $\Delta y_i^o$ as



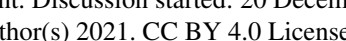



$$\Delta y_i^o = \frac{\frac{1}{(\sigma^p)^2}\bar{y}^p + \frac{1}{(\sigma^p)^2}y^o}{\frac{1}{(\sigma^p)^2} + \frac{1}{(\sigma^o)^2}} + \frac{\Delta y_i^p}{\sqrt{1 + (\frac{\sigma^p}{\sigma^o})^2}} - y_i^p, \quad (1)$$

where $i$ is the ensemble index, $y$ represents the observable state variable, and $\sigma$ is the error standard deviation. The superscript $p(o)$ always denotes the prior quantity estimated by the model (i.e., the observational quantity), and the overbar denotes the ensemble mean. The first term on the right hand is the adjusted ensemble mean, and the second term with the third term on the right hand is called the adjusted ensemble spread.

The second step regresses the observational increment onto the relevant model grids, which is computed by the first step. That
can be expressed as

$$\Delta x_i^u = \frac{cov(\Delta x, \Delta y)}{\sigma_y^2} \Delta y_i^o, \quad (2)$$

where $cov(\Delta x, \Delta y)$ is the covariance and $\sigma_y$ is the standard deviation, which are both evaluated by the model ensemble. Once all observations are looped over all the relevant model variables on the model grids, the analysis step is completed, and the model is initialized for the integration of the next step. For more details, please refer to Zhang and Rosati (2010a).

**2.3 Twin Experiment Design**

The implementation of CDA is a complex multi-task problem, which involves many factors, such as the coupled model bias, sampling of the observation system and verification of the analysis scheme. The uncertainty in any of these aspects may make the evaluation of a CDA system very difficult. In order to reduce the uncertainty and the evaluation complexity, a perfect model framework based on pseudo-observations (i.e., the perfect twin experiment) is adopted here to eliminate the influence
of model bias and observation system sampling on the assessment of the CDA system. The advantage of the perfect twin experiment design is that the known "truth" can be used as an accurate and reliable reference in evaluating the analysis quality of the CDA system. Similar framework has been used in Browne and Leeuwen (2015) to assess the performance of the equivalent weights filter within a coupled ocean–atmosphere general circulation model. In addition, when new assimilation components or observation types are added into the CDA system, the change of the assimilation performance can be effectively
quantified.

In the perfect twin experiment design, a time series of single-member model states are taken as the "truth", which can start from an arbitrary date (e.g., denoted as January 1980 in this paper). White noise is then added onto the "truth" to generate the pseudo-observations assimilated in the analysis stage. Thus, the assimilated observations here are gridded data at the grid points as the model variables. The added white noise is a parameter that determines the intensity of the observational constraint.
It is used to account for the random measurement error of the observation system, but does not include the representation error reflecting the limitation of the sampling scale. To sufficiently decorrelate the "truth" and the free integration of the model ensemble (i.e., the control experiment without assimilation of any observations), we use the restart of a 20-model-year free integration as the initial condition of the experiments.





The observation assimilated by ADA is surface pressure ($P_s$), and the standard deviation of the added observation error is 10
hPa; the observations assimilated by ODA are sea surface temperature (SST), three-dimensional temperature and salinity from
*in situ* ocean profiles which are generated from the "truth". The corresponding standard deviations of observation errors are
0.5 K for temperature and 0.1 PSU for salinity at the sea surface (typical error levels for SST and sea surface salinity) and
exponentially decay to one tenth of the surface values at 2000 m depth. These standard deviations have been trialed and adopted
in the previous similar studies (e.g., Zhang et al., 2007). The *in situ* ocean profiles are sampled using the real locations of Argo
data in 2007.

The ADA assimilation frequency is 6 hours and the ODA is 1 day. As an initial verification of the CESM-ECDA system, we
did not directly use a high-resolution experimental setup, but used a standard resolution ⌈100 km in ocean + 200 (100) km in
FV (SE) atmosphere⌉. Following previous studies (Zhang et al., 2005 & 2007) on covariance localization technique, and under
computational resource constraint, trials and errors are used to determine the ensemble size to be 12 in this perfect model study.
And the ensembles of initial conditions are constructed using the atmosphere states of 12 consecutive days.

Referring to the 20-CR (Twentieth Century Global Reanalysis, Compo et al., 2011), CFSR (NCEP Climate Forecast System
Reanalysis, Saha et al., 2010) and CM2.1-ECDA (Zhang et al., 2007), the horizontal localization radius is 2000 km in FV core
and 2500 km in SE core in ADA in this work. Considering that Ps represents the atmosphere weight of the whole air column
above the ground, $P_s$ observations are not only allowed to constrain $P_s$ state, but also used to update the pressure thicknesses
of all the alterable model levels. The analysis increment of $P_s$ is weighted and projected onto the pressure thicknesses of the
unsteady layers. Therefore, although $P_s$ itself is a two-dimensional observation, its physical property makes it contain abundant
three-dimensional information of the atmosphere, which can be transferred to the whole three-dimensional atmosphere by
updating the pressure thicknesses of the above model layers. The similar method has been successfully used in the 20-CR,
which assimilates only surface pressures using an ensemble Kalman filter method (Compo et al., 2011).
Referring to Zhang et al. (2007) and Kaspeck et al. (2018), the horizontal localization radius of SST observations is set to be
1000 km in ODA. In the vertical direction, SST observations are allowed to affect the ocean temperature of the 10 topmost
model levels (100 m depth). The horizontal localization radius of profiles temperature is 1000 km and the salinity is 500 km.
In addition, the horizontal correlation scale is multiplied by a $\cos(\theta)$ ($\theta$ is the grid latitude) factor up to 80°N (S) to make the
scale consistent with the characteristics of the Rossby deformation radius for a global analysis scheme (for more details, please
185   see Zhang et al., 2005 & 2007). Each observation is only allowed to impact at most two neighboring levels (one above and
one below), and the deepest profiles layer corrects the model values of all layers below.

The design of the evaluation experiments for the ADA, ODA and CDA is shown in Table 1. Given that our main purpose is to
document the development of the online CDA system for the community rather than provide reanalysis products, and the
constraint of computational resource, we only ran one model year to verify the reliability of the algorithm in this perfect twin
190   experiment. We can see that the DA in atmosphere and upper ocean has sufficiently converged. The ensemble experiments
include the control experiment (ctl) without assimilating any observations, the ADA experiment which only assimilates the $P_s$
observations with CAM-FV (ada_fv) and CAM-SE (ada_se), the ODA experiment assimilating only the SST observations



(oda_sst) and both the SST and *in situ* ocean profiles of temperature and salinity (TS profile; oda), and the CDA experiment assimilating both the $P_s$ and SST observations with CAM-FV (cda). The experiments' component setting in this study is BHISTC5 in FV and B1850CN in SE.

The validation of assimilation results is based on the root mean square error (RMSE), which is the most widely used statistic in system evaluation. RMSE is the standard deviation of the prediction errors, which measures how much the simulated data differ from the reference data (i.e., the "truth"). Since both atmosphere and ocean observations assimilated in this paper are located near the air-sea interface, and in principle, CDA can make more effective use of the observations near the interface and thus improve the coupled state estimation there, this paper mainly analyzes the model variables and fluxes near the interface. The atmosphere model variables near the interface used in this paper include the atmosphere surface pressure ($P_s$), surface temperature ($T_s$), surface wind components ($U_s$ and $V_s$) and surface specific humidity ($Q_s$). The analyzed ocean model variables include SST and ocean subsurface temperature and salinity. The fluxes at the air-sea interface used here include the water vapor flux (QF), the sensible heat flux (SHF), and the latent heat flux (LHF).

## 3 Development of Online DA Components of CESM

This section describes the framework and implementation of the online DA components of the CESM-ECDA system, which include the ADA component with CAM and the ODA component within POP. The ADA components are implemented to both CAM-FV and CAM-SE.

### 3.1 ADA with CAM-FV

#### 3.1.1 Online Ensemble Collection-distribution with CAM-FV Data Structure

CAM-FV adopts regular latitude-longitude grid in the horizontal direction, and both the model and assimilation are performed in parallel modes. The parallel domain decomposition in the model integration space is realized by dividing the global field in the horizontal direction based on the adjacent geophysical location. During the forecast stage, each processing element (PE) is responsible for a sub-domain of the global field of a single ensemble member, and different ensemble members are completely independent when the model integrated forward. When the model ensemble reaches the analysis time, a "super-parallel" technique (Zhang et al., 2007) is used to transmit the data between the model space and the analysis space. The super-parallel technique allows to make full use of the available computing resources and makes the model integration and the analysis be carried out online in an iterative manner.

Online data interaction based on the super-parallel technique mainly includes two stages: online ensemble collection and online ensemble distribution. Online collection refers to the transformation of the parallel domain decomposition and corresponding data storage form of the model space to those of the analysis space. As a result, each PE can obtain the ensemble data required by the ensemble filtering algorithm. On the contrary, online distribution indicates the transformation of the domain decomposition and data storage form of the analysis space back to those of the model space. In this way, the updated analysis





ensemble can be used as the initial conditions for the model integration of the next assimilation cycle. The transformation

between the model space and the analysis space is mainly based on the data collection and distribution functions of the Message

Passing Interface (MPI; Gropp et al., 1996), and it is an online data interaction mode via the memory-based reading/writing.

An online coupling strategy and implementing standards using MPI into an ensemble data assimilation system has been

detailed in Browne and Wilson (2015).

**Figure 1-a** is an example of the parallel domain decomposition and online collection and distribution with a total of 16 PEs

and 4 ensemble members with CAM-FV. In order to conduct the parallel task decomposition in the analysis space, the global

field is divided onto K PEs of the global PE list. To some extent, the parallel decomposition in the analysis space is arbitrary,

while **Figure 1-a** just shows a global domain decomposition according to a layout with $4 \times 4$ PEs. It should be noted that we

can also specify other decomposition layout types, such as $2 \times 8$ PEs etc. The halo is 12 grids in our experiments. In theory,

the halo should depend on the localization scale. But practically, we choose an appropriate halo according to some previous

experiments and model resolutions.

### 3.1.2 Implementation of Sequential EAKF Algorithm with CAM-FV

The implementation of the sequential EAKF with CAM-FV uses an approximate algorithm of the compute-domain/data-

domain strategy of Anderson (Anderson, 2001) to parallelize the filter. **Figure 2** shows a schematic of the implementation of

the sequential EAKF algorithm with CAM-FV. The algorithm assumes that the observations will only impact the "nearby"

model grids. Here "nearby" is generally defined by two conditions. One is that the nearby model grids must be located on the

current PE; the other is that the use of localization scheme requires that the nearby model grids should be within the localization

radius of the current observation. In the analysis space, the global field is divided into a group of analysis core domains (i.e.,

compute domain, see **Figure 1-a**) in the horizontal direction. Each core domain is surrounded by a certain number of nearby

grids, which are referred as halo (see **Figure 2**). An analysis core domain and its halo jointly constitutes an analysis domain

(i.e., data domain). The analysis process is based on the analysis domain, and each PE is responsible for the assimilation of

one analysis domain. When a set of observations are available, the online collection process transforms the required subset of

ensemble model states for each analysis domain to the corresponding PE. In each analysis domain, all available observations

are assimilated one by one using the EAKF algorithm (see Section 2.2) sequentially.

The ADA system with CAM-FV is realized using the two-step method of EAKF (Anderson, 2003; Zhang & Rosati, 2010)

based on the online ensemble collection and distribution processes. As **Figure 1-a** shows, after the online ensemble collection,

each PE obtains the ensemble vector of model states for an analysis domain. Then, the observational increment is calculated

based on Equation (1) on all PEs in parallel. It should be noted that one observation will be assimilated only if the ensemble

of all state variables required for the forward operator calculation is available on the current PE. After the observational

increments are obtained, they are projected onto the nearby model states to get the analysis increments via linear regression

expressed by Equation (2). Thus, the nearby model states can be updated by this observation via adding the analysis increments

onto the background model states. Therefore, the model states used in the assimilation of the current observation have already




been updated by all previous assimilated observations. This two-step assimilation process is repeated sequentially for the subsequent observations until all available observations of the current analysis step are processed. Finally, the analysis ensemble in the analysis core domains needs to be converted back to each member model space by online ensemble distribution

to be ready for the next integration stage.

### 3.2 ADA with CAM-SE

#### 3.2.1 Online Ensemble Collection-distribution with CAM-SE Data Structure

Unlike the regular latitude-longitude horizontal grid used in CAM-FV, CAM-SE uses a cubed-sphere grid in the horizontal direction (Dennis et al., 2012; Evans et al., 2013), which is no longer a logically rectangular grid. The direct effect of using

such a grid on the form of data storage in the model space is that the model states are no longer represented as two-dimensional variables in the horizontal direction, but are combined into one dimension. Taking the ne30np4 resolution of CAM-SE as an example, the two horizontal dimensions of the model variables in the geophysical space are represented by one dimension of length 48602 in the array form. Moreover, due to the characteristics of the cubed-sphere grid, two adjacent grid points in the horizontal one-dimensional representation of the variable with CAM-SE may not represent that the two grid points are adjacent

in the geophysical space. Therefore, the parallel decomposition strategy of the analysis domain with CAM-FV is no longer applicable to CAM-SE. Because the adjacent model grid points in the geophysical space can no longer be always ensured to be divided and stored on the same PE.

In order to implement the online ensemble collection and distribution with CAM-SE, the model state variables are vectorized. When the model reaches the analysis time, the ensemble of model states is first obtained based on the collection function of

MPI. This process is similar to that with CAM-FV, except that the collected state variables are all one dimension less than the same variables with CAM-FV. Then, all state variables of a single ensemble member are converted into a one-dimensional array. In principle, the arrangement of the model grid points in this array can be in an arbitrary order, because the location information of each point will be recorded by a separate array. In this way, the ensemble of all model state variables will be represented by a two-dimensional array, where one dimension corresponds to the grid point sequence number and the other

represents the ensemble sequence number. Then, the parallel decomposition of the analysis domain is achieved by dividing the grid point dimension over all PEs. As a result, each PE obtains the information about the ensemble of a subsequence of model state variables. However, it should be emphasized that the grid points in this subsequence in principle can be arbitrarily distributed and are not required to be adjacent in the geophysical space. At this point, the data conversion from the model integration space based on the online ensemble collection to the analysis space with CAM-SE is completed. When the

assimilation of all available observations at the current time is completed, the updated analysis ensemble obtained in the analysis space are converted back to the model space. This process is an inverse of the collection procedure and is implemented based on the MPI distribution function. **Figure 1-b** shows an example of the parallel domain decomposition and online collection and distribution with a total of 8 PEs and 4 ensemble members with CAM-SE.





### 3.2.2 Implementation of Parallel EAKF Algorithm with CAM-SE

The implementation of the ensemble filter with CAM-SE is also based on the two-step EAKF algorithm. Different from CAM-FV, the implementation needs to be adapted to the specific decomposition strategy of the analysis domains with CAM-SE. The differences are mainly in the computation of the observation prior ensemble and the regression of the observational increments. **Figure 3** shows the schematic of the implementation of the parallel EAKF algorithm with CAM-SE.

  The computation of the observation prior ensemble with CAM-SE uses the same method as DART (Anderson et al., 2009),
which is implemented based on the remote memory access (RMA) technique of MPI2 (Gropp et al., 1999). Because the grid points are divided onto different PEs via an arbitrary way, the four grid points enclosing one observation for interpolation may be located on different PEs from the owner PE of the observation. In the best case, all four enclosing points are located on the same PE as the owner PE; while in the worst case, all four enclosing points are located on four different PEs. Therefore, it needs to obtain the model background from the memories of other PEs (i.e., remote memories) for computing the observation
priors. In order to fulfillment this requirement, the RMA technique of MPI2 is used, which allows one PE to read or write asynchronously the memories of other PEs through a virtual window. Thus, regardless of which PEs the four enclosing points of the observation are located on, the prior ensemble for this observation can be obtained with aids of the RMA technique. Then the observational increment ensemble can be calculated based on Equation (1).

  The regression of the observational increments with CAM-SE is based on a parallel implementation of the EAKF algorithm
(Anderson & Collins, 2007). With CAM-FV, the observational increments are mapped onto the nearby model grids via linear regression to obtain the analysis increments of the nearby model states. With CAM-SE, the calculation of the observation priors requires access to the remote memories through RMA, which is more complex and computationally expensive than the direct access to the local memory with CAM-FV. To optimize the filtering algorithm with CAM-SE, the computation of the forward observation operator is implemented based on a parallel algorithm of Anderson and Collins (2007). This algorithm is
suitable for the parallel implementation of the EAKF algorithm. It splits the traditional forward observation operator computation, executed once for each observation, into a one-time calculation for all observations and an update of the nearby subsequent observation priors which have not been assimilated. Therefore, with CAM-SE, the forward operator computation is executed only once for each assimilation step. In this parallel implementation, the prior ensembles are first computed for all observations using the model background that has not been affected by any observation. Then, when an observation comes in,
its observational increments update not only the nearby model states as in CAM-FV, but also the prior ensembles of all nearby observations that have not been assimilated (i.e., the subsequent observation priors). This parallel algorithm has been shown to produce identical results to the traditional sequential algorithm. More details of this parallel implementation can be found in Anderson and Collins (2007).





### 3.3 ODA with POP

The online collection and distribution processes with POP are similar to those with CAM-FV (**Figure 1-a**). Although the horizontal grid used in POP is different from the regular latitude-longitude grid in CAM-FV, they both belong to the logically rectangular grid. More specifically, the horizontal dimensions of model variables can be represented by latitude and longitude. Therefore, similar parallel domain decomposition strategy based on the geophysical space to that with CAM-FV is used with POP to obtain the analysis domain. With CAM-FV, the decomposition of the analysis domains is based on the global horizontal

field. While with POP, the analysis domain decomposition is further optimized via a so-called local secondary decomposition on the model integration domain. In this local secondary decomposition strategy, the same model integration domains of all ensemble members are directly decomposed onto the PEs responsible for the integration calculation of these integration domains of all ensemble members. Taking **Figure 1-a** as an example, the southwest (i.e., lower-left) integration domains of 4 ensemble members are decomposed onto PEs 0, 4, 8, and 12 to obtain the analysis domains. Because the collection of the

entire global field of model states can be avoided (replaced by the collection of a subset of the global field), thus this local secondary decomposition strategy with POP reduces the hard limit on the PE storage capability. However, it should be pointed out that one disadvantage of this decomposition method is that the halo width in the analysis domain cannot exceed that of the integration domain. The implementation of the sequential EAKF algorithm with POP is the same as that of CAM-FV (see **Figure 2**), thus not repeated here.

### 3.4 The Online CESM-ECDA System

When the DA systems in CAM and POP have been developed respectively, the ocean-atmosphere CESM-ECDA system can be constructed. **Figure 1-c** shows the implementation framework of the CESM-ECDA system. In one assimilation cycle, the execution of CESM-ECDA system can be described as follows. The CESM model reads in the initial condition ensemble to start the forward integration of the model ensemble. When the model integration reaches the observation time, the model

ensemble integration is suspended. Then the forecast fields of CAM and/or POP are obtained as the background fields of the assimilation by the ADA and/or ODA components through the online ensemble collection, then the two-step update of EAKF is used for the sequential assimilation of the observations in each component. After the assimilation process has been completed, the analysis fields obtained by ADA and/or ODA are transformed back to corresponding model spaces through the online ensemble distribution. Then the analysis ensemble updated by the observations is used as the initial states for the model

ensemble to continue the integration in the next forecast stage. It should be noted that the ADA and ODA components can be executed using the same or different frequencies. Generally, the ODA interval is longer than that of ADA to account for the different characteristic time scales between the ocean and the atmosphere. Because the background fields used in ADA and ODA come from the atmosphere and ocean components of the coupled CESM, and the dynamic coupling between the atmosphere and ocean components is performed through the interface fluxes in the model forecast stage. The observed

information in the atmosphere and ocean can be exchanged with each other, so that the coupled state estimation obtained by





the CESM-ECDA is more self-consistent and balanced. In addition, because the coupled covariance between the atmosphere and the ocean is not used in the current CESM-ECDA system, the observation in one component is not allowed to directly update the model state in the other component in the assimilation stage, which makes it a weakly coupled DA (WCDA) system.

## 4 The Evaluation of CESM-ECDA System with prefer twin experiment

### 4.1 ADA with CAM-FV

The assimilation of $P_s$ observations by ADA with CAM-FV significantly improves the atmospheric surface variables. **Figure 4 (a-e)** shows the time series of RMSEs of five atmospheric surface variables and Table 2 shows the global averaged RMSEs. To focus on the impact of assimilation after the system reaches equilibrium, the globally averaged RMSEs of atmospheric variables and fluxes are calculated with the experiment output data of the first month excluded in this paper. By assimilating

$P_s$ observations, not only the surface pressure, but also other atmospheric variables are significantly improved. Compared with the ctl, the RMSEs of surface variables are clearly and rapidly reduced by assimilating $P_s$ observations. The global averaged RMSE of $P_s$ is reduced from 6.68 hPa to 4.05 hPa, nearly improved by 40%. The RMSEs of other four variables are reduced by about 15%–25%.

**Figure 5 (a-e)** shows the distribution of the ada_fv-to-ctl (hereafter ada_fv/ctl) RMSE ratio of the five atmospheric surface

variables. Compared to ctl, which does not assimilate observation, the assimilation of $P_s$ in ada_fv can significantly reduce the RMSE of $P_s$ all over the globe. The RMSEs of $U_s$ and $V_s$ are also reduced throughout the globe in ada_fv. Though the RMSEs of $T_s$ and $Q_s$ are increased in some regions in ada_fv, the assimilation of $P_s$ observations can improve the analysis accuracy over most regions. Besides, the improvements are mostly located at the mid-latitudes on both hemispheres, especially in the south hemisphere. $P_s$ is a two-dimensional variable, but it contains abundant three-dimensional information of the atmosphere.

Therefore, the solo assimilation of $P_s$ observations not only significantly corrects the model $P_s$ field, but also improves other atmospheric variables. It should be pointed out that U, V, T and Q are not used as direct assimilating variables, but are adjusted through the dynamic process and physical process of the model after assimilating $P_s$. And the errors of these variables are also significantly reduced, which shows that the assimilation effect of the system in the model is reasonable. The conclusion is consistent with previous studies such as 20-CR (Compo et al., 2011).

### 4.2 ADA with CAM-SE

The atmospheric surface variables by assimilating the $P_s$ observations within CAM-SE are also significantly improved. **Figure 6 (a-e)** shows the time series of RMSEs of five atmospheric surface variables and Table 3 shows the global averaged RMSEs. The calculation of the global averaged RMSEs also excludes the output data of the first month. Generally speaking, the assimilation effects of $P_s$ observations with CAM-FV and CAM-SE are similar. The ADA system with CAM-SE can also

significantly improve the atmospheric variables over the ctl experiment, even only the $P_s$ observations are assimilated. Furthermore, the amplitude of the RMSE reduction with CAM-SE tends to be smaller than that with CAM-FV, especially for





P$_s$, U$_s$ and V$_s$ (Table 2 vs. Table 3). For example, compared with the ctl experiment, ada_fv reduces the RMSEs of P$_s$, U$_s$, and V$_s$ by up to 39.4%, 25.5%, and 26.1%, respectively, while the corresponding RMSE reductions in ada_se are 29.4%, 20.1%, and 17.0%, respectively. In addition, the horizontal distribution of RMSE in SE (not shown here) is similar to that in FV.

**Figure 7 (a-f)** shows the time series of RMSEs of upper layer atmospheric variables, being temperature and winds at 870 hPa and 510 hPa as examples. The RMSEs of all these three variabilities in ada_se are also smaller than that in ctl experiment in high-level atmosphere. At 870 hPa level, the RMSEs of temperature, zonal wind and meridional wind from ctl are 3.43 K, 6.18 m/s and 5.45 m/s. After assimilating the only surface pressure, these RMSEs are decreased to 3.02 K, 4.99 m/s and 4.64 m/s respectively. At 510 hPa level, the RMSEs from ctl are 3.32 K, 8.72 m/s and 8.01 m/s, while the RMSEs from ada_se are

3.00 K, 7.72 m/s and 7.24 m/s respectively. The model errors of other atmospheric variables are also reduced, which will not be displayed one by one here.

### 4.3 ODA

#### 4.3.1 Assimilation of SST

The assimilation of SST observations by the ODA system with POP significantly improves the accuracy of the ocean states.

**Figure 8 (a-b)** shows the time series of RMSE of SST from ctl and oda_sst, and the distribution of the oda_sst-to-ctl (hereafter oda_sst/ctl) RMSE ratio of SST. To focus on the impact of assimilation after the ocean system reaches equilibrium, the global averaged RMSE and the RMSE ratio of the oceanic variables are computed with the output data in which the first three months were excluded in this paper. Compared with the ctl experiment, the RMSE of SST in oda_sst significantly and quickly reduces at the beginning of the experiment, and then gradually decreases further and stays stably. The SST RMSE in oda_sst reduces

from 0.58 K to 0.13 K. The distribution of the RMSE ratio (oda_sst/ctl) shows that the oda_sst experiment can improve the quality of SST over almost the entire globe, except in the high-latitude regions of the north hemisphere. Besides, the distribution of the improvement in oda_sst is relatively uniform compared with that in the ADA experiments discussed above. By assimilating SST, the RMSE of SST is significantly reduced, which is consistent with previous studies such as CFSR (Saha et al., 2010).

#### 4.3.2 Assimilation of *in situ* Ocean Profiles

The design of the assimilation of *in situ* ocean profile temperature and salinity observations with POP is similar to the EAKF algorithm of CM2.1-ECDA (Zhang et al., 2007). Since the locations of the real profile observations are changing over time and they are not coincident with the model grid points. The first step of profile assimilation is to get the model values at the positions of profiles by interpolating and then calculating the observational increment. When calculating the observational

increment, eight-points interpolation is used, that is, both the upper and lower four points are used for the interpolation to obtain the model value at the observational point. The second step is to project the observational increment onto the surrounding model grid points. In the vertical direction, each profile affects one layer above and one layer below, with a total





of two model layers. Same as the previous study (Zhang et al., 2007), the impact from temperature to salinity and the impact from salinity to temperature are both activated. In accordance with the SST assimilation, the frequency of profile observations

assimilation is also 1 day. The distributions of the profile observations are shown in **Figure 9 (a-d)**.

The addition of ocean profile observations to the ODA system further significantly improves the ocean state estimates over the control run. **Figure 10 (a-d)** shows the time series of RMSEs of ocean temperature and salinity vertically averaged from 0 – 500 m and 500 – 2000 m from the ctl and oda experiments. And Table 4 shows the global averaged RMSEs of these four variables, with the data of the last six months. Compared with the ctl experiment, the RMSEs of temperature and salinity are

both significantly reduced. Assimilation of temperature and salinity profile observations rapidly reduces the RMSE compared to ctl at the beginning of the experiments in 0 – 500 m. Compared with the salinity, the RMSE of temperature decreases more rapidly to a stable level (approximately 3 – 4 months for the temperature vs. 7 – 8 months for the salinity). Besides, at the same depth range, the improvement to the temperature tends to be larger than the salinity, with the RMSE reduction of 25.0% (9.9%) and 7.1% (1.2%) for the 0 – 500 m (500 – 2000 m) temperature and salinity, respectively. And for the same variable, the

reduction of RMSE in the shallower ocean is larger than the deeper ocean (Table 4), which may be caused by the slower variability of the deeper ocean.

Compared with ctl, oda greatly improves the ocean temperature and salinity estimates, especially in the shallow ocean (0 – 500 m). The RMSEs of ocean temperature and salinity are clearly reduced in most regions of the globe between the surface and 500 m depth, in addition to the coast of Africa and the high latitude areas of the North Pacific. The largely improved

regions are mostly located in the low- to mid-latitude regions, especially in the Pacific and Indian Oceans. In the deeper ocean (500 – 2000 m), the improvement of temperature and salinity is less significant than in the upper ocean, showing a small RMSE reduction from ctl compared to the upper ocean. Overall, the improvement of salinity is smaller than temperature. This method of improving ocean model state estimate by assimilating temperature and salinity profiles has also been applied in previous studies, such as Zhang et al. (2007) and Carton et al. (2018). The conclusion in this work is basically consistent with them.

**4.4 CDA**

After the ADA and ODA components have been implemented based on CAM and POP, respectively, the CESM-ECDA system is also constructed. Because the ocean-atmosphere coupled error covariance is not used in the CESM-ECDA system, the observation in one component system (such as the atmosphere) cannot directly affect the model state in another component (such as the ocean) in the analysis stage. Therefore, the CESM-ECDA system is implemented in the WCDA style. In the

current version of the CESM-ECDA system, the ADA component is capable of assimilating the atmospheric observations of the two-dimensional surface pressure, the three-dimensional temperature, wind components and humidity; and the ODA component can assimilate the oceanic observations of the two-dimensional SST and the three-dimensional *in situ* profiles of temperature and salinity. Besides, the localization, covariance inflation and incremental analysis update (IAU; Bloom et al., 1996) schemes are also included. More specifically, the localization schemes include the variable localization, horizontal

localization and vertical localization based on the widely-used filter from Gaspari and Cohn (1999), and the covariance





### 4.4.1 Impact of CDA on Atmosphere State Estimation

The CESM-ECDA system can improve the quality of the atmosphere state estimation over the single ADA experiment. When
the assimilation of $P_s$ observations in the atmosphere and SST observations in the ocean are both activated, the CDA system
operates in a WCDA style (i.e., the cda experiment). Compared with the single ADA experiment, the cda experiment can be
used to evaluate the assimilation performance of the CESM-ECDA system in the atmosphere. **Figures 11 (a-e)** show the time
series of RMSEs of five atmospheric surface variables and SST. When the $P_s$ and SST observations are both assimilated, the
cda experiment can further reduce the RMSEs of the atmospheric variables in comparison with the ada_fv experiment, which
only assimilates the $P_s$ observations into the atmosphere.

Table 5 lists the global averaged RMSEs of these five variables and three important air-sea interface fluxes with the data.
Although the cda experiment can improve all the atmospheric surface variables considered here over ada_fv, the error
reductions are small (approximately $1\% - 2\%$, and the $T_s$ is 4.3%). This may be caused by the strong correlation between the
ocean temperature and atmosphere temperature near the air-sea interface, while the correlation between SST and other
atmospheric surface variables is weak. Compared with ada_fv, the cda experiment further includes the assimilation of SST
observations into the ocean component. Therefore, the model SST state is well constrained by the SST observations. The
improved SST in cda can further benefit the overlying atmosphere through the ocean-atmosphere dynamic coupling. Although
all atmospheric states should be improved by the better lower boundary conditions provided by the ocean, only those strongly
correlated with the SST more significantly benefit from the improved SST.

The cda experiment shows a mixed distribution of decreased and increased RMSE and a global averaged weak improvement
for $P_s$, $Q_s$, $U_s$, and $V_s$. The regions of significant improvement are mostly located in the tropical eastern Indian Ocean and the
tropical to subtropical Atlantic, which may indicate the more significant positive impact of the coupling between SST and air-
sea fluxes. It should be noted here that the cda experiment can significantly improve the $T_s$ state over almost the entire ocean
areas, especially in the tropical Indian Ocean, the tropical to subtropical Atlantic, and the mid-latitude Pacific.

The improvement of CESM-ECDA over single ADA experiment can also be reflected in the air-sea interface fluxes. **Figure
12 (a-f)** shows the time series and ratio (cda/ada_fv) distribution of RMSEs of QF, SHF, and LHF. These fluxes can also be
improved by the further assimilation of the SST observations into the ocean in the cda experiment. Compared with ada_fv, cda
further reduces the RMSEs of QF, SHF, and LHF by 2.1%, 3.8%, and 2.1% (Table 5), respectively. These three fluxes share
significant improved regions with above variables, such as the tropical Indian Ocean and the tropical to subtropical Atlantic.
The assimilation of SST observations in cda improves the air-sea coupling processes in these regions, leading to corrected
interface fluxes there. Then the improved interface fluxes further transmit the correcting information to the overlying
atmosphere.





### 4.4.2 Impact of CDA on ocean State Estimation

The CESM-ECDA system can obtain improved ocean states over the single ODA experiment. **Figure 11-f** shows the time
series of the SST RMSE from the oda_sst and cda experiments. Compared with oda_sst which only assimilates the SST
observations, cda further assimilates the $P_s$ observations into the atmosphere component. Therefore, the comparison between
oda_sst and cda can be used to evaluate the impact of the CESM-ECDA system on the ocean state estimation. Assimilation of
SST observations alone in the oda_sst experiment has already largely reduced the SST RMSE in comparison with the ctl
experiment. While cda can further reduce the SST RMSE by up to 10.4% (from 0.134 K to 0.120 K, with the data of the first
three months excluded from the computation). The RMSE of SST can be significantly reduced in most regions except in high-
latitude areas by further assimilating the $P_s$ observations into the atmosphere. The improved areas are mostly located in the
broad mid-latitude regions of the two hemispheres, especially in the South Pacific and the adjacent South Ocean. This may be
because that the air-sea coupling in the mid-latitude regions is dominated by the driving effect of the atmosphere to the
underlying ocean. When $P_s$ observations are assimilated into the atmosphere, the atmospheric states are better constrained,
which provides an improved upper boundary conditions for the underlying ocean. In the mid-latitude regions, the atmosphere
has a significant driving effect on the ocean, thus it is more conducive to the transmission of the observational information
from the overlying atmosphere to the underlying ocean.

### 5 The real observation assimilating experiment

The results of the perfect twin experiment show that the CESM-ECDA system can work well. To further verify its performance
in the real world, a three-year reanalysis experiment (hereafter referred to as real-CDA) from 1978 to 1980 (one-year ODA
and two-year CDA) with the component setting of BHISTC5 (historical run with CAM5) is conducted using 12 ensemble
members. Surface pressure from ERA-interim (Dee et al., 2011) and grided observational SST from HADISST (Rayner et al.
2003) and three-dimensional temperature and salinity profiles from XBT, CTD, MBT and OSD are assimilated. Considering
that observations below 2000 m are very sparse and internal variability in the deep ocean is weak, we employ global restoring
of climatological temperature and salinity (e.g. Levitus 2001; 2005; 2012) in the deep ocean to relax the distorted ocean
stratification caused by strong data constraint in upper ocean (Lu et al., 2020). A control experiment (real-CTL) is also
conducted with the same setup except that it does not employ data assimilation.

We compare the PS and SST results of real-CDA with those of CFSR (Saha et al., 2010), 20CR (only for PS, Compo et al.,
2011) and ERA-20C (Poli et al., 2013) in **Figure 13 (a-d)** for 1980.The RMSE of Ps in real-CDA (red line) is smaller than the
CTL (black line) and 20CR (yellow) but lager than the CFSR (purple line) and ERA-20C (pink line). The mean RMSEs of
CTL, real-CDA, CFSR, 20CR and ERA-20C are 15.57, 11.94, 9.21, 13.89 and 6.68 hPa, respectively. In these reanalysis
products, the 20CR and ERA-20C are similar to the real-CDA experiment in this study, which assimilates mainly Ps in
atmosphere. The CFSR assimilates all kinds of observations, so the result of CFSR is better than real-CDA. It is worth to





mention that the ERA-20C and ERA-interim are both form ECMWF using similar atmospheric models/schemes. It is
reasonable that the ERA-20C has the best result because we use ERA-interim as the observation.

The RMSE of SST in real-CDA is smaller than real-CTL (black) and ERA-20R (pink line) and close to CFSR (purple line).
The mean RMSEs are 1.41, 0.16, 0.16, and 0.56 °C in real-CTL, real-CDA, CFSR and ERA-20C, respectively. From the
distributions (Fig 13-c,d), we see that the Ps RMSE in real-CDA decrease 30%~50% in the global world compared to CTL in
addition to Africa. Compared with ERA-20C, the SST RMSE is greatly reduced worldwide to an extend about 30% ~ 90%.

**6 Summary and Discussions**

In this paper, an online ensemble atmosphere-ocean coupled data assimilation system within the Community Earth System
Model, which consists of the ODA component and the ADA components (both for the finite-volume and the spectral-element
dynamical cores for ADA), is designed and evaluated systematically. In the ADA component, the surface pressure observations
are assimilated. In the ODA component, the gridded SST observations and the *in-situ* profiles of temperature and salinity are
both assimilated. The perfect twin experiments have been conducted steadily for one model year for the single ADA
components, the single ODA component, and the weakly coupled CESM-ECDA system. The results show that the CESM-
ECDA system is effective in assimilating observations in both atmosphere and ocean. By assimilating the surface pressure
observations alone, the RMSEs are significantly reduced by up to 39% for $P_s$ and 16% – 26% for $T_s$, $Q_s$, $U_s$ and $V_s$ within
CAM-FV. The RMSE reductions for the single ADA experiment within CAM-SE are generally smaller but still significant
than those in CAM-FV. By assimilating the SST observations alone, the SST RMSE is greatly reduced by up to 77%. When
the three-dimensional *in situ* temperature and salinity profiles are further assimilated in the ODA system, the three-dimensional
ocean temperature and salinity can be significantly improved.

When the atmospheric and oceanic observations are jointly assimilated, the CESM-ECDA system executes in a WCDA style.
Results show that the CESM-ECDA system can obtain robustly improved state estimations in both atmosphere and ocean
compared with the corresponding single component DA experiments discussed above, respectively, which confirms the
stability and effectiveness of the established CESM-ECDA system. The atmosphere-ocean coupled assimilation allows to
make more effective use of the available observations in both atmosphere and ocean components. Therefore, the observational
information in different component systems is allowed to be transmitted and exchanged across the air-sea interface in the
forecast stage of the coupled model. It is worth mentioning that due to the significant difference of the characteristic spatial
and temporal scales between the atmosphere and the ocean and the difference of the air-sea coupling mechanisms in different
regions, the obtained coupled state estimation shows a more complicated distribution.

Furthermore, the reanalysis experiment with real observations shows that the Ps RMSE of CESM-ECDA is smaller than 20CR
if we take ERA-interim as truth. The SST RMSE of CESM-ECDA is smaller than ERA-20C and close to CFSR when
HADISST is assimilated.





A previous adaptively inflated ensemble filter study has been designed to enhance the consistency of upper- and deep-ocean adjustments, which is based on "climatological'" standard deviations being adaptively updated by observations (Zhang et al., 2010a). But in this study, we only focus on the system design and evaluation of the CESM-ECDA, the inflation is not used in this study. The inflation scheme can be considered into the ADA system in the next work. While in ocean, the situation is more complex because the variability is different in upper- and deep-ocean. It needs much deeper and further work to explore the

inflation in this CESM-ECDA system.

In this study, our purpose is to document the design and evaluation of the online CESM-ECDA system instead of providing a climate reanalysis product. Only several one-year perfect twin experiments and a three-year analysis experiment with real observation are conducted. In follow-up studies, we plan to complete a half century climate reanalysis within an improved CDA system using EnOI-Like filtering (Yu et al., 2019) which can effectively solve the problem of redistribution between

model integration and assimilation, in which only one dynamical model member is integrated.

The online CESM-ECDA system reported in this study uses multiple ensemble members to carry out the assimilation process. The required computing resources are large for such an ensemble-based CDA system with the full complexity CGCM to execute, especially when using a high-resolution configuration (e.g., the CESM-HR, Zhang et al., 2020a). Besides, observations of other types and variables need to be further assimilated into the system. In addition, the current CESM-ECDA

system is only a weakly coupled system in which the observations in one component cannot directly update the state in another component. It is worth mentioning that some ocean dynamic processes (particularly in the Tropics, such as ENSO) have influences over long spatial scale. However, depending on the reliability of statistical correlation, such as the ensemble size, the localization radius is a tunable parameter in practical data assimilation. Considering the compressibility and that the Rossby radius of deformation in atmosphere is general larger than ocean, the impact radius of atmosphere is larger than ocean in this

work. We highlight here that an appropriate utilization of localization is important in the real data assimilation and coupled reanalysis, and more studies shall be further explored.

It is worth mentioning that a high-resolution configuration of CESM has been performed and evaluated thoroughly for century-long climate simulations (Small et al., 2014; Zhang et al., 2020a), and participated in the High-Resolution Model Intercomparison Project (HighResMIP, Roberts et al., 2020). A 500-year preindustrial control simulation and a 250-year

historical and future climate simulation have also been completed and evaluated (Zhang et al., 2020a; Chang et al., 2020). In this paper, the CESM-ECDA system is only evaluated with a standard resolution. In the future, the CESM-ECDA system will be assessed with the high-resolution version of CESM. Referring to the multi-timescale EnOI-like high-efficiency approximate filter (Yu et al., 2019), our ultimate goal is to develop a computationally efficient CDA system suitable for the CESM-HR (e.g., 25-km ADA system within CAM-SE and 10-km ODA system within POP2) using only one CESM integration instead

of multiple ensemble members. This CDA system with CESM-HR is expected to be used to produce a high-resolution coupled reanalysis with CESM using various real observations. Therefore, many work remain to be done in the future. The ability to assimilate real observations of various types needs to be included in the future, by which similarities and differences with the previous studies (e.g., Zhang et al., 2007; Lu et al., 2020) can be compared and the important climate phenomena such as El





Niño–Southern Oscillation (ENSO) and Atlantic meridional overturning circulation (AMOC) can also be further explored. In
addition, the cross-domain coupled error covariance is desired to be included in the future to extend the current weakly CDA
system to a strongly one.

## 6 Code availability

The codes of CESM-ECDA System are available at ZENODO via https://doi.org/10.5281/zenodo.5733849.

## 7 Data availability

The data used in this work is available at ZENODO via https://doi.org/10.5281/zenodo.5733849.

## 8 Author contributions

SZ, WZ, JS, and YJ initiated this research, proposed most ideas, and co-led paper writing.

SJ, YJ led the process of development & research and contributed equally to this work.

All authors contributed to the improvement of ideas, software testing, experimental evaluation, and paper writing/proofreading.

## 585 9 Competing interests

The authors declare that they have no conflict of interest.

## Acknowledgments

The research is supported by the Chinese NFS projects 41830964,41775100, 311021009 and the Key R & D program of
Ministry of Science and Technology of China (2017YFC1404100, 2017YFC1404104, 2018YFC1406202), as well as
Shandong Province's "Taishan" Scientist Program (ts201712017) and Qingdao "Creative and Initiative" frontier Scientist
Program (19-3-2-7-zhc).

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

## Figures:


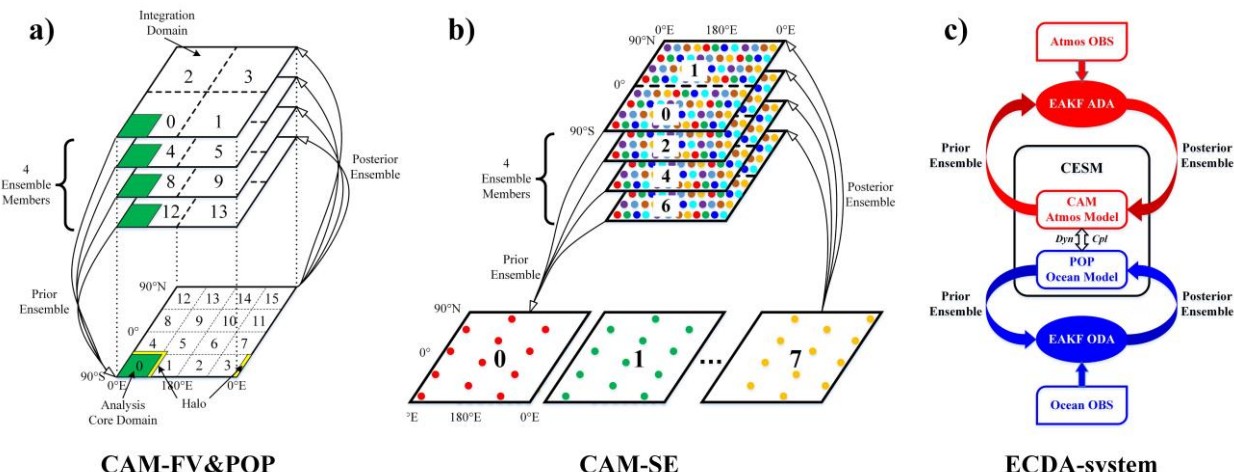

**Figure 1. a) The parallel domain decomposition and the online ensemble collection-distribution of a scalar field with CAM-FV. The four ensemble members are integrated forward in time in parallel, using four processors each member. At analysis time, the ensemble members are synchronized and the global four-element ensemble vectors are decomposed onto all 16 analysis processors. For each physical field, each analysis processor sequentially uses the observations to update the ensemble vectors at each grid point in its core domain (green) and halo (yellow). Once all nearby observations have been assimilated, the updated ensemble vectors in the core domains are transmitted back to the integration processors, completing the cycle.**

**b) The parallel domain decomposition and the online ensemble collection-distribution of a scalar field with CAM-SE. Different from CAM-FV, CAM-SE uses the cubed-sphere grid. To address this grid change, the implementation of domain decomposition with CAM-SE adopts a similar fashion to that used in DART (Anderson et al., 2009). This new method allows to "randomly" assign model states to different PEs. And the calculation of forward operator is realized based on the MPI2 remote memory access (RMA). At analysis time, the ensemble members are synchronized and the global ensemble vectors are randomly decomposed onto all 8 PEs (grid points with same color in the figure).**

**c) The parallel domain decomposition and the online ensemble collection-distribution of a scalar field with CAM-SE. Different from CAM-FV, CAM-SE uses the cubed-sphere grid. To address this grid change, the implementation of domain decomposition with CAM-SE adopts a similar fashion to that used in DART (Anderson et al., 2009). This new method allows to "randomly" assign model states to different PEs. And the calculation of forward operator is realized based on the MPI2 remote memory access (RMA). At analysis time, the ensemble members are synchronized and the global ensemble vectors are randomly decomposed onto all 8 PEs (grid points with same color in the figure).**









**Figure 2. Schematic of the implementation of the sequential EAKF algorithm with CAM-FV. The implementation is based on the two-step method of EAKF. The decomposition strategy allows adjacent grids to be divided into the same analysis domain (i.e., onto the same analysis PE). One analysis core domain (compute domain) and its halo constitute one analysis domain (data domain). When the observations at a given time are available, only the prior ensemble of a single observation is first computed. Then each observation is assimilated sequentially according to the two-step EAKF algorithm. The observational increments of an observation only regress on the nearby model states.**




**Figure 3. Schematic of the implementation of the parallel EAKF algorithm with CAM-SE. The implementation is also based on the**
**two-step method of EAKF. In CAM-SE, the grid points are decomposed onto the analysis PEs via an arbitrary way. Therefore, the**
**computation of the forward observation operator may need to obtain values from other PEs, which is realized through the MPI2**
**RMA technique. The lower-right dashed rectangular illustrates an example of grabbing data from PEs 1 – 3 to PE0 via a virtual**
**window of RMA. When the observations at a given time are available, the prior ensembles for all observations are computed. The**
**observational increments of an observation regress not only on the nearby model states, but also on the subsequent observation prior**
**ensembles.**





**Figure 4. Time series of RMSEs of (a) surface pressure Ps, (b) surface temperature Ts, (c) surface specific humidity Qs, (d) surface zonal wind Us, and (e) surface meridional wind Vs from ctl and ada_fv.**






**Figure 5.** The distribution of the RMSE ratio (ada_fv/ctl) of (a) surface pressure Ps, (b) surface temperature Ts, (c) surface specific humidity Qs, (d) surface zonal wind Us, and (e) surface meridional wind Vs. The RMSE ratios are calculated with the data of the first month excluded.








**Figure 6. Time series of RMSEs of (a) surface pressure Ps, (b) surface temperature Ts, (c) surface specific humidity Qs, (d) surface zonal wind Us, and (e) surface meridional wind Vs from ctl and ada_se.**







**Figure 7. Time series of RMSEs of (a) 870 hPa temperature, (b) 510 hPa temperature, (c) 870 hPa zonal wind, (d) 510 hPa zonal wind, (e) 870 hPa meridional wind and (f) 510 hPa meridional wind from ctl (black lines) and ada_se (green lines).**




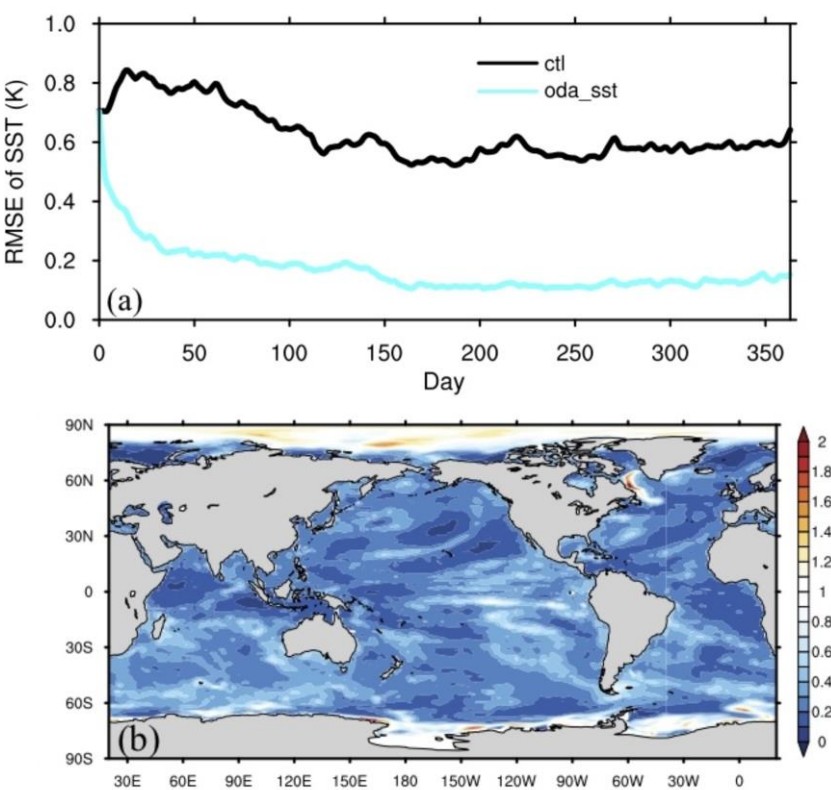

**Figure 8. a) Time series of RMSE of SST from ctl and oda_sst, and b) the distribution of the RMSE ratio (oda_sst/ctl) of SST. The RMSE ratio is calculated with the data of the first three months excluded.**





**a)2007-01-01**

**b)2007-07-01~2007-07-10**

**c)2007-01**

**d)2007**

**Figure 9. Argo locations on a) 2007-01-01, b)2007-07-01~07-10, c) 2007-Jan and d) 2007.**



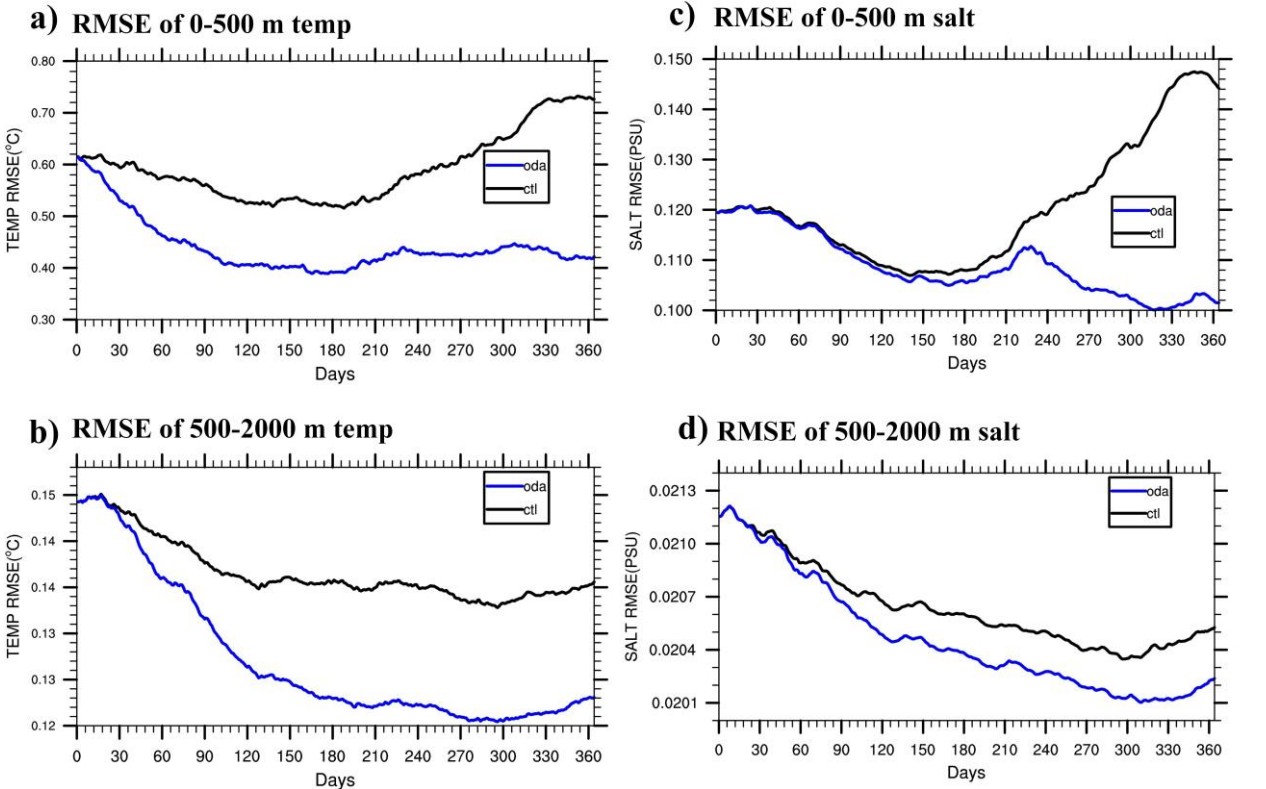

Figure 10. Time series of RMSEs of a) 0-500 m, b) 500-2000 m ocean temperature and c) 0-500 m, d) 500-2000 m ocean salinity from ctl (black line) and oda (blue line).





**Figure 11. Time series of RMSEs of (a) surface pressure Ps, (b) surface temperature Ts, (c) surface specific humidity Qs, (d) surface zonal wind Us, (e) surface meridional wind Vs and (f) SST from ada_fv/oda and cda.**



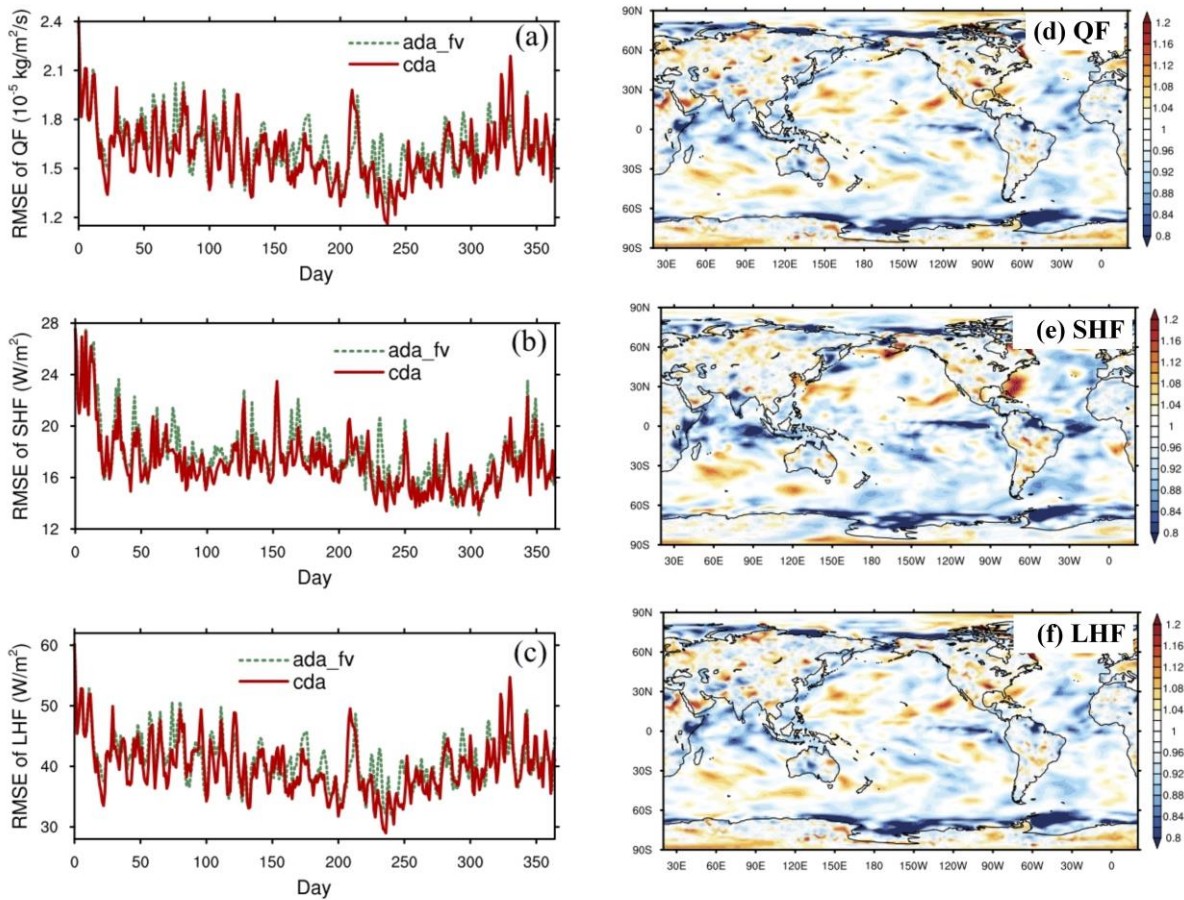

**Figure 12. Time series of RMSEs of (a) water vapor flux QF, (b) sensible heat flux SHF, and (c) latent heat flux LHF from ada_fv and cda, the distribution of the RMSE ratio (cda/ada_fv) of (d) water vapor flux QF, (e) sensible heat flux SHF, and (f) latent heat flux LHF. The RMSE ratios are calculated with the data of the first month excluded.**





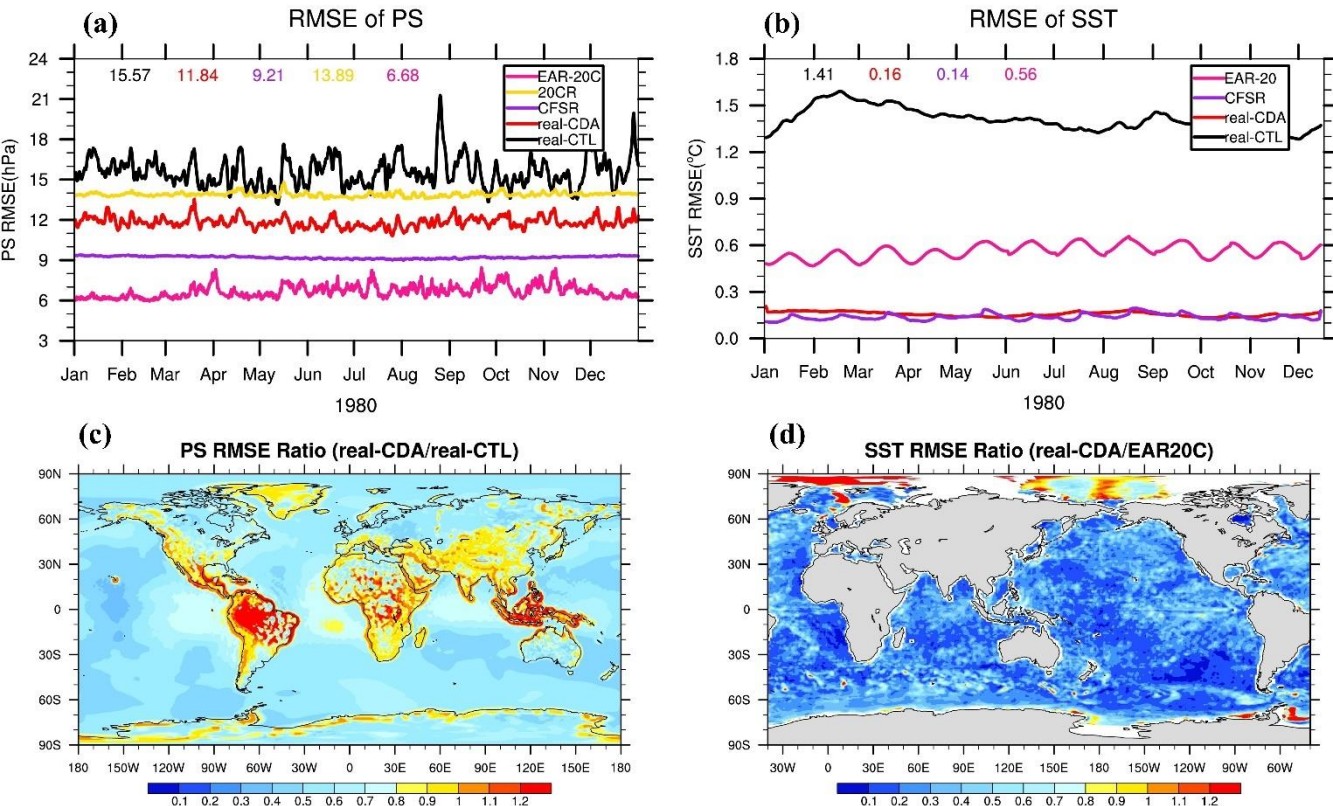

**Figure 13. Time series of RMSE of a) Ps and b) SST from the real observation assimilating experiment and RMSE ratio distributions of Ps (real-CDA/real-CTL) and SST (real-CDA/EAR20C) in 1980. The number in a a/b are RMSEs of the mean value corresponding to each product.**

## Tables:

**Table 1. List of experiments.**

| Experiment | DA | OBS |
|---|---|---|
| ctl | none | none |
| ada_fv | CAM-FV ADA | Ps |
| ada_se | CAM-SE ADA | Ps |
| oda_sst | POP ODA | SST |
| oda | POP ODA | SST + TS profile |
| cda | CAM-FV ADA + POP ODA | Ps + SST |

**Table 2. Globally averaged RMSEs of surface pressure Ps, surface temperature Ts, surface specific humidity Qs, surface zonal wind Us, and surface meridional wind Vs from ctl and ada_fv. The RMSEs are calculated with the data of the first month excluded.**

| Experiment | Ps (hPa) | Ts (K) | Qs (g/kg) | Us (m/s) | Vs (m/s) |
|---|---|---|---|---|---|





| | | | | | |
|---|---|---|---|---|---|
| ctl | 6.68 | 2.99 | 1.64 | 4.24 | 4.36 |
| ada_fv | 4.05 | 2.51 | 1.39 | 3.16 | 3.22 |
| reduction(%) | 39.4 | 16.1 | 15.5 | 25.5 | 26.1 |

**Table 3. Globally averaged RMSEs of surface pressure Ps, surface temperature Ts, surface specific humidity Qs, surface zonal wind Us, and surface meridional wind Vs from ctl and ada_se. The RMSEs are calculated with the data of the first month excluded.**

| Experiment | Ps (hPa) | Ts (K) | Qs (g/kg) | Us (m/s) | Vs (m/s) |
|---|---|---|---|---|---|
| ctl | 6.17 | 2.47 | 1.58 | 4.62 | 4.49 |
| ada_se | 4.36 | 2.10 | 1.32 | 3.69 | 3.72 |
| reduction(%) | 29.4 | 14.9 | 16.2 | 20.1 | 17.0 |

855

**Table 4. Globally averaged RMSEs of ocean temperature and salinity vertically averaged between surface and 500 m and between 500 m and 2000 m from ctl and oda with the data of the last six months.**

| Experiment | 0-500m T (K) | 500-2000m T (K) | 0-500m S (PSU) | 500-2000m S (PSU) |
|---|---|---|---|---|
| ctl | 0.59 | 0.14 | 0.121 | 0.0204 |
| oda | 0.44 | 0.13 | 0.109 | 0.0202 |
| reduction(%) | 25.0 | 7.1 | 9.9 | 1.2 |

860 **Table 5. Globally averaged RMSEs of surface pressure Ps, surface temperature Ts, surface specific humidity Qs, surface zonal wind Us, surface meridional wind Vs, water vapor flux QF, sensible heat flux SHF, and latent heat flux LHF from ada_fv and cda. The RMSEs are calculated with the data of the first month excluded.**

| Experiment | Ps (hPa) | Ts (K) | Qs (g/kg) | Us (m/s) | Vs (m/s) | QF (10-5kg/m2/s) | SHF (W/m2) | LHF (W/m2) |
|---|---|---|---|---|---|---|---|---|
| ada_fv | 4.05 | 2.51 | 1.39 | 3.16 | 3.22 | 1.62 | 17.53 | 40.44 |
| cda | 3.98 | 2.40 | 1.35 | 3.12 | 3.19 | 1.58 | 16.86 | 39.60 |
| reduction(%) | 1.8 | 4.3 | 2.4 | 1.3 | 1.1 | 2.1 | 3.8 | 2.1 |