# Peer review of "An Online Ensemble Coupled Data Assimilation Capability for the Community Earth System Model: System Design and Evaluation"

_Geoscientific Model Development, 2021_

## Author Response (AR1)

**Reviewer #1**

This work developed a weakly coupling assimilation system for Community Earth System Model (CESM) using EAKF. This is demonstrated by OSSE twin experiments. Different from some assimilation systems of CESM developed by other groups using the DART, an assimilation software tool developed by NCAR, this work develops a technique for on-line assimilation analysis using the compute-domain/datadomain data process strategy and parallel schemes. The authors claim such an on-line assimilation scheme can save a lot computational expense compared with commonly used off-line assimilation by DART. Indeed, the computational expense is a critical issue in numerical simulation and prediction, especially for real-time operational prediction. Thus, I support this work although it only addresses the technique advantage, not much about the scientific merits of data assimilation itself. However, I have two large concerns and wish the authors to address them before I recommend it to be accepted.

In general:
The author would like to thank the reviewer for the thorough examination and comments on this manuscript. All the suggestions and comments from the reviewer are very helpful to improve presentation of the manuscript. After plenties of discussions among co-authors, we have carefully revised the manuscript according to reviewer's comments and suggestions.

The following is the point-by-point reply to the comments:

1) The most spotlight of this work is based on the conclusion that the computational expense of such an on-line assimilation system is much computationally economic and efficient, compared with the traditional off-line system. However, this work does not show any evidence to support this conclusion. The authors should present details and results on the computational efficiency, for example, CPU time for one step of assimilation, the entire time of an period of analysis etc.. A systematical comparison against the off-line experiment is the most idealized and expected. Without these supportive evidences, this work seems empty and pale.
**RE**: A good suggestion! We have added more details and discussions on the computational efficiency as the new **Section 4.5** in the revision. Please see **Table 6** and **Lines 546-555**.

2) The readers of this work should mainly be geoscientists, who may not be strong in computer science. The authors should consider the scope of readers in presenting their work so that this work can be fully understood and further reproduced by the readers of interests. Thus, I suggest authors to pay attention to the presentation in describing how to implement on-line the assimilation system.
**RE**: Thank you very much for your insightful suggestion. More descriptions on how to implement online assimilation system are added in the revision. Please see **Lines 530-545**.

**Reviewer #2**

This manuscript documented the development of an ensemble coupled data assimilation (ECDA) capability for the community earth system model (CESM). A traditional ocean data assimilation (ODA) was used in the CESM_ECDA, while the atmosphere data assimilation (ADA) part just assimilates surface pressure data. The CESM-ECDA was evaluated using one set of perfect data assimilation experiment and another set of real observation assimilation experiment.

The development of ECDA for CESM is an important work for CESM's climate prediction capability. One novelty of this work to only use surface pressure data in ECDA's ADA part. However, the implemented algorithm of the surface pressure data assimilation, which is different from the literature, needs further clarification and scientific justification. Therefore, I recommend "major revision" of this manuscript.

In general:

The authors would like to thank the reviewer for careful and meticulous examinations on this manuscript. All the suggestions and comments from the reviewer help improve presentation of the manuscript very much. After the discussions among co-authors, we have carefully revised the manuscript according to reviewer's comments and suggestions.

The following is the point-by-point response for the specific comments.

Specific comments:

1, For the surface pressure data assimilation, there are two important references updated recently. One is that the 20-CR has been updated to version 3 (20CRv3, Slivinski et al., 2019); Another is that a similar surface pressure data assimilation has been used in GFDL's SPEAR coupled model (Yang et al. 2021).

Slivinski, L. C., Compo, G. P., Whitaker, J. S., Sardeshmukh, P. D., Giese., B. S., McColl, C., et al. (2019) Towards a more reliable historical reanalysis: Improvements for version 3 of the Twentieth Century Reanalysis system. Q. J. R. Meteorol. Soc., 145, 2876-2908. https://doi.org/10.1002/qj.3598

Yang, X., T. L. Delworth, F. Zeng, L. Zhang, W. F. Cooke, M. J. Harrison, A. Rosati, S. Underwood, G. P. Compo, C. McColl, 2021: On the Development of GFDL's decadal prediction system: initialization approaches and retrospective forecast assessment, Journal of Advances in Modeling Earth Systems, 13, e2021MS002529

**RE**: Thanks to the reviewer's guidance. These two important references are added in the revision. Please see **Lines 169-170**.

2, P6L175-180, In this study, the analysis increment is weighted and projected onto the pressure thickness at each model layer. This algorithm is different from 20-CR (Compo et al., 2011; Slivinski et al, 2019) and Yang et al. (2021), in which the analysis increments of winds, temperature and moisture are directed solved via the covariance with surface pressure. Without the simultaneous

increments of winds, temperature and moisture, the dynamical balance between those fields and surface pressure would not be maintained. The authors need provide further justification for the choice of this algorithm.

RE: Thanks for the insightful comment about the algorithm. More discussions and further justification about this algorithm are added into the revision, please see **Lines 159-178**. We also provide further results of the middle troposphere and the upper troposphere. Please see **Fig. 7, Fig.8 and Fig. 9**.

3, P12L360, RMSE is a useful metric for assessing the data assimilation performance. One aspect of surface pressure data assimilation shown in 20CR is that 20CR could have very similar weather-to-climate scale variability in the troposphere as other traditional atmosphere reanalyses which use all available observations. Therefore, it is important to assess how well the weather variability in the middle troposphere (e.g., variability of the daily 500-mb geopotential heights) is retrieved by the surface pressure data assimilation.

RE: Good suggestion! We have provided the results of variability of the 6 hourly 500-mb geopotential heights. The results of 500-mb geopotential height are improved by assimilating only surface pressure. Please see **Fig. 8a~b** and **Fig. 9a**, and **Lines 410-424**.

4, P12L370, "U, V, T and Q are not used as direct assimilation variables, ….. The conclusion is consistent with previous studies such as 20-CR (Compo et al., 2011)." Further evidence is needed to support this conclusion, since the dynamical balance between U, V, T and Q and Ps is not maintained in the data assimilation step without the direct increments of U, V, T and Q. Some plots of representing weather variability in the upper troposphere (e.g., figure 7 and 9 in Compo et al, 2011) would be useful.

RE: Good suggestion! We have provided some plots of the 6 hourly 300-mb geopotential heights which are referred to figure 7 and 9 in Compo et al, 2011. The results of 300-mb geopotential height are also improved by assimilating only surface pressure. Please see **Fig. 8c~d** and **Fig. 9b**, and **Lines 425-433**.

5, P13L385-390. The reduction rate of RMSEs for the variables in the upper layers is much smaller than that in the surface. This might suggest that the simultaneous increments of U, V, T and Q are very important for representing the atmosphere state in the whole troposphere.

RE: Agree! Discussions have been added in the revision. Please see **Lines 405-409**. Thanks.

6, P15L470-475, Are QF and LHF the same variable in physics just with different unit? Fig. 12d and f looks exactly the same.

RE: Yes. QF is water vapor flux which is calculated from LHF, and LHF is latent heat flux. We retain the LHF and removed the QF. Please see **Fig 14**. Thanks!

7, P12L485, It is worth showing the global plot of the SST improving due to CDA.

RE: A good point! Please see **Fig 14-f** and **Lines 526-529**.

8, P16L500-505. 20CRv3 should be used for this comparison.

**RE**: A good suggestion! The result of 20CRv3 has been used for comparison. Please see **Fig. 15** and **Lines 567, 570-571**.

9, P17L510. ERA-20C data is an atmosphere reanalysis product. I don't see any scientific merit of comparing SST from ERA-20C, since the SST should be observed HADISST. The SST RMSE reduction compared with ERA-20C is simply due to the difference between different SST observations.

**RE**: Thanks for the insightful comment. We replaced ERA-20C with the result of CERA-20C which is a coupled reanalysis. Please see **Fig. 15** and **Lines 572-579**.

---

## Author Response (AR2)

**Reviewer #1**

In general:
The author would like to thank the reviewer for the thorough examination and comments on this manuscript.
The following is the point-by-point reply to the comments:

Specific comments:

1, In the revised Figure 15, the Ps RMSE of CESM-ECDA is smaller than 20CRv2 but is larger than 20CRv3 (Line 569-571), so it is not fair to state that Ps RMSE is smaller than 20CR. Several statements relevant to this point has not been updated yet.

a. L29-30. "Results show that Ps RMSE is smaller than 20CR and SST RMSE is better ERA-20C and close to CFSR." I would suggest removing this statement from the abstract.

**RE**: The statement has been removed. Thanks!

b. L604-605. "Furthermore, the reanalysis experiment with real observations shows that the Ps RMSE of CESM-ECDA is smaller than 20CR if we take ERA-interim as truth."

**RE**: The statement has been updated. Please see **Lines 603-604**. Thanks to point out!

2. The two references (Slivinski et al. 2019; Yang et al. 2021) were not added in the reference list.

**RE**: These two references have been added in the reference list. Please see **Lines 799-801 and Lines 819-821**. Thank you very much!